# Phenotypic and Genomic Comparison of *Klebsiella pneumoniae* Lytic Phages: vB_KpnM-VAC66 and vB_KpnM-VAC13

**DOI:** 10.3390/v14010006

**Published:** 2021-12-21

**Authors:** Olga Pacios, Laura Fernández-García, Inés Bleriot, Lucia Blasco, Antón Ambroa, María López, Concha Ortiz-Cartagena, Felipe Fernández Cuenca, Jesús Oteo-Iglesias, Álvaro Pascual, Luis Martínez-Martínez, Pilar Domingo-Calap, María Tomás

**Affiliations:** 1Microbiology Department-Research Institute Biomedical A Coruña (INIBIC), Hospital A Coruña (CHUAC), University of A Coruña (UDC), 15006 A Coruna, Spain; olgapacios776@gmail.com (O.P.); laugemis@gmail.com (L.F.-G.); bleriot.ines@gmail.com (I.B.); luciablasco@gmail.com (L.B.); anton17@mundo-r.com (A.A.); maria.lopez.diaz@sergas.es (M.L.); cortizcartagena@hotmail.com (C.O.-C.); 2Study Group on Mechanisms of Action and Resistance to Antimicrobials (GEMARA), Spanish Society of Infectious Diseases and Clinical Microbiology (SEIMC), 28003 Madrid, Spain; felipefc@us.es (F.F.C.); jesus.oteo@isciii.es (J.O.-I.); apascual@us.es (Á.P.); luis.martinez.martinez.sspa@juntadeandalucia.es (L.M.-M.); 3Spanish Network for Research in Infectious Diseases (REIPI), 41071 Sevilla, Spain; 4UGC Enfermedades Infecciosas, Microbiología Clínica y Medicina Preventiva, Instituto de Biomedicina de Sevilla (IBIS), Hospital Universitario Virgen Macarena/CSIC/Universidad de Sevilla, 41071 Sevilla, Spain; 5National Centre for Microbiology, Institute of Health Carlos III, 28029 Madrid, Spain; 6Departamento de Microbiología, UGC Microbiología, Hospital Reina Sofía, Instituto Maimónides de Investigación Biomédica de Córdoba (IMIBIC), Universidad de Córdoba, 14014 Cordova, Spain; 7Institute for Integrative Systems Biology (I2SysBio), Universitat de València-CSIC, 46980 Paterna, Valencia, Spain; pilar.domingo@uv.es

**Keywords:** lytic phages, *Klebsiella pneumoniae*, genomic annotation, homing endonucleases, L-shaped tail fiber

## Abstract

*Klebsiella pneumoniae* is a human pathogen that worsens the prognosis of many immunocompromised patients. Here, we annotated and compared the genomes of two lytic phages that infect clinical strains of *K. pneumoniae* (vB_KpnM-VAC13 and vB_KpnM-VAC66) and phenotypically characterized vB_KpnM-VAC66 (time of adsorption of 12 min, burst size of 31.49 ± 0.61 PFU/infected cell, and a host range of 20.8% of the tested strains). Transmission electronic microscopy showed that vB_KpnM-VAC66 belongs to the *Myoviridae* family. The genomic analysis of the phage vB_KpnM-VAC66 revealed that its genome encoded 289 proteins. When compared to the genome of vB_KpnM-VAC13, they showed a nucleotide similarity of 97.56%, with a 93% of query cover, and the phylogenetic study performed with other *Tevenvirinae* phages showed a close common ancestor. However, there were 21 coding sequences which differed. Interestingly, the main differences were that vB_KpnM-VAC66 encoded 10 more homing endonucleases than vB_KpnM-VAC13, and that the nucleotidic and amino-acid sequences of the L-shaped tail fiber protein were highly dissimilar, leading to different three-dimensional protein predictions. Both phages differed significantly in their host range. These viruses may be useful in the development of alternative therapies to antibiotics or as a co-therapy increasing its antimicrobial potential, especially when addressing multidrug resistant (MDR) pathogens.

## 1. Introduction

Bacteriophages are the most abundant organisms in the biosphere, and the natural predators of bacteria, their unique host [1,2]. In the actual context of increasing antibiotic resistance towards nearly all the antibiotics used in the clinical setting, therapeutic alternatives to antibiotics are welcomed, and the clinical use of bacteriophages (or phages), known as phage therapy, is included among these. This strategy is defined as the administration of lytic phages, which display a lytic cycle of infection, directly to a patient with the purpose of lysing (and thus killing) the bacterial pathogen that is causing a clinically relevant infection [3,4]. Although the importance of phage therapy was reduced with the advent and golden age of antibiotics, it has recently experienced a well-deserved renewal of interest [4]. The clinical trials that have been carried out using lytic phages against different human infections demonstrate favorable safety profiles and, in some cases, encouraging evidence of efficacy [4,5,6].

Interestingly, a common tendency currently is to combine phages with conventional antibiotics, with the hope of achieving a synergistic effect and reducing the arising of resistance [7,8]. In addition, the use of several phages combined into a cocktail preparation is also a common solution, which has revealed no unwanted side-effects, a decrease in the occurrence of resistance, and an expansion in the therapeutic spectrum [4].

The number of clinical strains that a particular bacteriophage can infect after recognizing their bacterial receptor in the host surfaces is called the host range. To adhere to the host receptors, phages use their receptor binding proteins (RBPs) or the receptor binding domains (RBDs) within their L-shaped tail fibers, which are proteins that recognize oligo-mannose units on the bacterial cell surface (i.e., lipopolysaccharide) [9]. It is well known that phages can modify the RBDs in their tail fibers to infect different hosts [10].

In this work, we isolated two lytic phages that infect clinical strains of *Klebsiella pneumoniae* from sewage water, extracted their DNA, and sequenced them by Illumina platform. The size of the genomes from both phages, vB_KpnM-VAC13 and vB_KpnM-VAC66, was included in the 100- to 200-kbp range [11] (large phages), and showed a high similarity between them, with the exception of the L-shaped tail fiber protein and the content of homing endonucleases: we found that vB_KpnM-VAC13 has a genome size of 178,532 pb, whereas the size of vB_KpnM-VAC66 is 174,826 pb. Therefore, we phenotypically characterized vB_KpnM-VAC66 (as vB_KpnM-VAC13 was already characterized in a previous work [12]), annotated its genome, and compared it with the genome of vB_KpnM-VAC13. We also performed a phylogenetic study and a host range assay to determine the number of strains among several clinical isolates of *K. pneumoniae* that they could infect. This work sheds some light on the possibility of treating multidrug resistant (MDR) isolates of this pathogen using effective lytic phages, as the phenotypic and genomic analysis that we report here constitutes an initial but necessary step in the use of lytic phages alone or in combination with conventional antibiotics (or other drugs) with which they may be synergized.

## 2. Materials and Methods

### 2.1. Bacterial Strains and Phages

We used 48 clinical strains of *K. pneumoniae*, which were obtained from the Virgin Macarena University Hospital (Spain) and the National Centre for Microbiology (Carlos III Health Institute, Madrid, Spain). The bacterial strains were cultured using Luria-Bertani broth (LB, containing 1% tryptone, 0.5% yeast extract and NaCl and 2% agar) and, for the experiments where phages were added, the medium was supplemented with 1 mM of CaCl_2_ to favorize the adsorption. In each case, strains were incubated at 37 °C with shaking (180 rpm).

### 2.2. Isolation and Propagation of Klebsiella Phage vB_KpnM-VAC66 and Stability of the Phage Titers at 4 °C over Time

vB_KpnM-VAC66 is a lytic bacteriophage of *K. pneumoniae* isolated from sewage water. For this purpose, 50 mL of sewage water were centrifuged for 10 min at 4000 rpm and the supernatant was filtered through 0.45 and 0.22 μm filters to remove cellular debris. Aliquots (1 mL) of the samples were added to 500 μL of different *K. pneumoniae* isolates by the double-agar layer method [13]. Plates were incubated at 37 °C for 20 h. Isolated lytic plaques of different morphology were recovered with a micropipette and stored at −80 °C. To purify the plaques, two additional plaque-picking steps were performed. The plaque-purified phages were amplified and propagated in liquid media, infecting their natural hosts when they reached an OD_600nm_ = 0.4. Lysed bacteria were removed by centrifugation and supernatants were stored at 4 °C. To assess the stability of bacteriophages over time at 4 °C, titration of the stocks of both bacteriophages dating from one year was performed by serially diluting and plating using the double-agar layer method in TA medium (containing 1% tryptone, 0.5% NaCl, and 1.5% agar).

### 2.3. Transmission Electron Microscopy (TEM)

A quantity of 600 µL of the lytic bacteriophage vB_KpnM-VAC66, diluted in SM buffer (100 mM NaCl, 10 mM MgSO_4_, 20 mM Tris-HCl pH 7.5) and carrying approximately 10^9^ plaque-forming units (PFU) per mL, was negatively stained with 1% aqueous uranyl acetate before examination under a transmission electron microscope (TEM JEOL 1011).

### 2.4. Host Range of vB_KpnM-VAC13 and vB_KpnM-VAC66 in the Collection of Clinical Strains of K. pneumoniae: Spot Test and Efficiency of Plating (EOP)

The host range of the lytic phages was established by performing the spot test and the efficiency of plating (EOP) assays. The spot test assay was undertaken as described by Raya et al. [14]: a drop of 15 µL of each bacteriophage was spotted over TA plates prepared by the double-agar layer method as previously mentioned, and including the clinical strain of interest [13]. The EOP assay was undertaken as previously described by Kutter [15]. The EOP value was calculated as the ratio between the phage titer (PFU/mL) of the test strain and the titer of the host strain (*K. pneumoniae* ATCC^®^10031^TM^ for vB_KpnM-VAC13 and the clinical isolate *K. pneumoniae* K3320 for vB_KpnM-VAC66). For both assays, TA-soft medium was supplemented with 1 mM of CaCl_2_ and used to make plates by the double-agar layer method [13]. Each strain was tested in triplicate, and SM buffer was included as negative control.

### 2.5. Adsorption Curve

To determine the time of adsorption of *Klebsiella* phage vB_KpnM-VAC66, an overnight culture of *K. pneumoniae* K3320 was diluted 1:100 in LB supplemented with 1 mM CaCl_2_ and incubated at 37 °C at 180 rpm. At OD_600nm_ = 0.3, the culture was infected with the lytic bacteriophage vB_KpnM-VAC66 at a multiplicity of infection (MOI) of 0.01. The culture was statically maintained at room temperature and 1 mL of culture medium was removed every 4 min for 20 min, serially diluted into SM buffer, and plated by the double-agar method in triplicate. The following day, PFUs were enumerated and the percentage of free phages was determined [16].

### 2.6. One-Step Growth Curve

To determine the latency period and the burst size of this bacteriophage, which is the average number of virions released per infected cell, a one-step growth curve was performed [16]. Briefly, an overnight culture of *K. pneumoniae* K3320 was diluted 1:100 in LB broth supplemented with 1 mM CaCl_2_ and incubated at 37 °C at 180 rpm until OD_600_ = 0.3; then, 1 mL of the culture was infected with vB_KpnM-VAC66 at a MOI of 0.01. It was maintained static at room temperature for the adsorption period, then centrifuged at 6000 rpm for 10 min and washed with LB medium. This step was repeated three times. Then, flasks containing 20 mL of LB supplemented with 1 mM CaCl_2_ were inoculated with 20 µL of this bacteria-phage mix and incubated at 37 °C and 180 rpm. Aliquots were taken at different time points (0, 5, 10, 15, 20, 30, and 40 min post-infection), serially diluted, and plated by the double-agar method. For every curve, data were analyzed using GraphPrism v.6.

### 2.7. Infection Curve

To assess the lytic capacity of vB_KpnM-VAC66, infection curves at different MOIs were performed in triplicate. An overnight culture of the clinical isolate *K. pneumoniae* K3320 was diluted 1:100 in LB broth supplemented with 1 mM CaCl_2_ then incubated at 37 °C at 180 rpm until the early logarithmic phase (OD_600nm_ = 0.3) was reached. At this moment, the lytic bacteriophage vB_KpnM-VAC66 was added to the culture at MOIs of 0.1 and 1, and OD_600nm_ was measured for 6 h at 1 h intervals. To assess if resistance arose, another measure was taken at 22 h post-infection.

### 2.8. Mutation Rate to Bacteriophage vB_KpnM-VAC66 in K. pneumoniae K3320 Clinical Strain

The frequency of resistant mutants was calculated as previously described [17]. Briefly, an overnight culture of the strain K3320 was diluted 1:100 in LB and grown to an OD_600nm_ of 0.7. An aliquot of 1 mL of the culture containing 10^8^ CFU/mL was serially diluted, and the corresponding dilutions were mixed with 50 μL of vB_KpnM-VAC66 at 10^9^ PFU/mL, then plated by the double-agar layer method in TA medium. The plates were incubated at 37 °C for 24 h, then the CFUs were enumerated. The mutation rate was calculated by dividing the number of resistant bacteria (growing in the presence of the bacteriophage) by the total number of bacteria plated in conventional LB agar.

### 2.9. Bacteriophage DNA Isolation

The phage DNA was extracted from purified phage particles using the phenol-chloroform-isoamyl alcohol (PCI) method and precipitated using ethanol, based on the following protocol: https://phagesdb.org/media/workflow/protocols/pdfs/PCI_SDS_DNA_Extraction_2.2013.pdf, accessed on 1 February 2021. The precipitated DNA was centrifuged, washed with 500 μL of 70% ethanol, re-centrifuged and, after removing the supernatant, the pellet was air-dried (10–20 min) and finally dissolved in 50 μL of dH_2_O. Quantification of DNA was measured by a NanoDrop ND1000 Spectrophotometer (ThermoFisher Scientific, Waltham, MA, USA) and Qubit fluorometer (ThermoFisher Scientific, Waltham, MA, USA).

### 2.10. Genome Sequencing, Analysis, and Annotation

Whole genome sequencing of phages was performed using the Illumina Nextseq 500 system. The sequenced raw data were processed to obtain high-quality reads, then assembly was performed using the PATRIC server and database (https://www.patricbrc.org/ accessed on 22 February 2021), particularly the SPAdes v3.15.2 software. Functional annotation of the predicted proteins was first assessed with the PATRIC server, which uses the RAST tool kit, then via the BLAST program (https://blast.ncbi.nlm.nih.gov/Blast.cgi accessed on 2 March 2021), HMMER (https://www.ebi.ac.uk/Tools/hmmer/search/phmmer accessed on 3 March 2021), and HHPRED (https://toolkit.tuebingen.mpg.de/tools/hhpred, accessed on 5 March 2021) online tools.

### 2.11. Phylogenetic Study

Twenty nucleotide sequences corresponding to complete genomes were downloaded from the GenBank database (NCBI) and aligned using the MAFFT server (https://mafft.cbrc.jp/alignment/server/index.html, accessed on 26 October 2021). Then, a phylogenetic tree was generated using the maximum likelihood method of MEGA X software (bootstrap of 100), which builds trees using branch swapping and evaluates them by a measure of compound probability. The confidence levels of each clade are shown.

## 3. Results

### 3.1. Phenotypic Characterization of vB_KpnM-VAC66

In a previous work, our group characterized the lytic bacteriophage vB_KpnM-VAC13 and enhanced its antimicrobial activity by combining it with imipenem and with a repurposed drug, mitomycin C, against carbapenemase-producing and imipenem persister *K. pneumoniae* isolates [12]. In this work, we sequenced and annotated its genome together with that belonging to vB_KpnM-VAC66. Before this, we performed a phenotypic characterization of vB_KpnM-VAC66.

#### 3.1.1. Morphological Characterization

When the lytic bacteriophage vB_KpnM-VAC66 (representing a virus of Bacteria, infecting *K. pneumoniae*, with myovirus morphology, named vB_KpnM-VAC66 [18]) was characterized, it was observed that it produced clear plaques of 1 mm of diameter in plates of TA supplemented with 1 mM CaCl_2_. The virion morphology was observed by TEM, which revealed the typical structure of a tailed bacteriophage belonging to the *Myoviridae* family, with an icosahedral capsid of approximatively 100 nm in length and 77 nm in width, and a rigid and contractile tail of around 107 nm long and 26 nm wide (Figure 1a).

#### 3.1.2. Host Range

The host range of each bacteriophage, vB_KpnM-VAC13 and vB_KpnM-VAC66, was determined by testing their infectivity in a collection of 48 clinical strains of *K. pneumoniae*. The results obtained by spot test showed that the phage vB_KpnM-VAC13 infected 66.7% and the phage vB_KpnM-VAC66 infected 62.5%. When these results were confirmed by EOP, a reduction in the infectivity of both phages was observed, showing that the host range covered 20.8% of the tested strains. In addition, it was noticeable that, despite being genetically very similar, their lytic spectrum was different in 42% of the strains with positive EOP (Table 1).

#### 3.1.3. Characterization of the vB_KpnM-VAC66 Infection

The adsorption and one-step growth curves revealed a time of adsorption of 12 min to its native host, the clinical isolate *K. pneumoniae* K3320 (Figure 1b), with a latent period of 5 min and a burst size (average of released virion particles) of 31.49 ± 0.61 PFU/infected cell (Figure 1c, denoted as L and B respectively).

The infection curve of vB_KpnM-VAC66 was derived using the clinical isolate K3320 and infecting it at MOIs of 0.1 and 1; we observed a higher lysis at MOI 1 compared to MOI 0.1 during the first 6 h of infection, with drastic decrease in the OD_600nm_ compared to the growth in the absence of phage (control). However, resistance increased for both conditions at 22 h post-infection (Figure 1d).

We determined the emergence of resistant mutants of *K. pneumoniae* K3320 to the lytic bacteriophage vB_KpnM-VAC66, and we found that this value was 5 × 10^−2^ ± 0.03. The mutation rate to vB_KpnM-VAC13 phage, previously calculated, was 3.96 × 10^−3^ for the clinical isolate *K. pneumoniae* K2534 (in which this bacteriophage was characterized [12]).

### 3.2. Genomic Analysis and Comparison of the Phages vB_KpnM-VAC13 and vB_KpnM-VAC66

The whole genome sequence of the phages vB_KpnM-VAC13 (GenBank: MZ322895, BioSample SAMN22059222) and vB_KpnM-VAC66 (GenBank: MZ612130, BioSample SAMN22059211) were obtained and analyzed, showing a size of 178,532 and 174,826 pb, respectively, encoding 286 and 289 coding sequences (CDS) (Figure 2). We found that the genomes were arranged in a particular disposition: no lysis-specific blocks were distinguished, and structural and morphogenesis-related proteins were repeated in various blocks along the genome (Figure 2). This type of genome, which is not very well organized into functional clusters but shows dispersed agglomerates, has previously been described for big phages of *K. pneumoniae*, also belonging to the *Tevenvirinae* subfamily, within which the T4 coliphage is the best studied [19].

The genomes of both phages were compared and they showed a nucleotide similarity of 97.56% and a query cover of 93% (Figure 3a). The dot plot performed with the ViPTree tool (https://www.genome.jp/viptree/, accessed on 27 October 2021), based on proteomic homologies, showed the high similarity between both genomes (Figure 3b). The comparison of the ORFs was undertaken and resulted in 87% of the ORFs exhibiting ≥95% identity; 5% exhibited between 85% and 95% identity; 2% of the ORFs exhibited <85% identity; and 6% of the CDS showed no homology at all (Figure 3c). Among the latter, there were 21 ORFs (that is, ≤10%) that differed (Table 2).

#### 3.2.1. Early Genes: DNA Replication and Transcription Regulation

##### DNA Replication Proteins

We observed many genes encoding proteins involved in the DNA replication, causing independence from the host [20]. The synthesis of DNA during replication of vB_KpnM-VAC13 and vB_KpnM-VAC66 starts from RNA primers made by the DNA primase, encoded by the ORF127 in vB_KpnM-VAC13 and ORF118 in VAC66. The ORFs 165 (vB_KpnM-VAC13) and 143 (vB_KpnM-VAC66) encode the RNase H, which is likely responsible for the removal of primers. This process requires single-strand DNA-binding (ssDNA) proteins (ORFs 161 and 139 in vB_KpnM-VAC13 and vB_KpnM-VAC66), which bind DNA behind RNase H [21]; the ssDNA proteins are also required during the elongation stage of DNA replication, which explains the identification of more ssDNA proteins, at least in vB_KpnM-VAC13 (ORFs 56 and 109). ORFs 209 (vB_KpnM-VAC13) and 192 (vB_KpnM-VAC66) encode DNA-ligase that ligates the Okazaki fragments and is also involved in repairing DNA mismatches and sealing gaps [20]. Other key proteins implicated in the DNA replication of these lytic phages are the numerous helicases and clamp loader subunits found in both genomes.

##### Transcription Regulation

Different transcription factors were found throughout the genomes of both phages, such as the transcription repressor of early genes in both sequences (ORF156 and 134 in vB_KpnM-VAC13 and vB_KpnM-VAC66, respectively), and the regulators of the late genes: the promoters encoded by ORF163 and 141, respectively, and the antitermination protein Q (ORF286 and ORF18), which allows the transcription of late genes.

#### 3.2.2. Late Genes: Virion Maturation, Lysis of the Host and Defense

##### Morphogenesis Proteins

Within this group, we included genes encoding structural components such as the head, prohead and capsid, the baseplate structure, the portal protein, the neck and whiskers proteins, and the tail fibers (Figure 2). Phages belonging to the *Myoviridae* family are unified by the presence of a tail-sheath protein, which forms part of the rigid tail typical of these viruses and contracts upon host infection, initiating viral DNA injection in the bacterial cell [22]; in this case, both vB_KpnM-VAC13 and vB_KpnM-VAC66 encode this protein (ORF47 and ORF22, respectively), and their nucleotidic sequences showed a 99% identity.

##### Lysis-Related Proteins

Furthermore, we found several lytic proteins spread in both genomes, such as: (i) lysin motif (LysM)-containing peptidoglycan binding protein (ORF133 in vB_KpnM-VAC13 and ORF68 in vB_KpnM-VAC66); (ii) inner (i-) and outer (o-) membrane spanins (ORFs 185 and 186 in vB_KpnM-VAC13, and 165 and 166 in vB_KpnM-VAC66), which constitute a complex that forms a disulfide-linked complex that spans the host inner and outer membranes, contributing to their disruption [23]; (iii) holin (ORF106 in vB_KpnM-VAC13 and ORF81 in vB_KpnM-VAC66), which presumably oligomerize to form holes in the cytoplasmic membrane to allow the passage of endolysins and, finally, (iv) two adjacent endolysins (ORFs 272 and 273 in vB_KpnM-VAC13 and 253 and 254 in vB_KpnM-VAC66), enzymes that hydrolyze the peptidoglycan [24]. We found that one among the two endolysins predicted for each bacteriophage (ORF272 for vB_KpnM-VAC13 and ORF253 for vB_KpnM-VAC66) encoded the endopeptidase domain PET-M15-4, which displays L-Ala-D-Glu activity and was found in several mycobacteriophages and in some phages infecting *Salmonella* or *Bacillus cereus* [25,26,27]. All these proteins were highly similar when compared between both phages (≥95% identity), except the i-spanin (an integral cytoplasmic membrane protein), for which the nucleotidic and amino acid sequences showed identities of 87% and 86%, respectively.

##### Interference with Bacterial Metabolism and Defense

As shown in Figure 2, a large proportion of DNA-related proteins are encoded by these two genomes. We found the presence of ADP-ribosyltransferases (ORFs 73 and 48 in vB_KpnM-VAC13 and in vB_KpnM-VAC66, respectively), which are thought to help in hijacking some bacterial networks, as they modify the α-subunit of the bacterial RNA polymerase (RNAP) to make it switch preference for viral templates [28]. We also observed the presence of methyltransferases in both genomes (ORF145 in vB_KpnM-VAC13 and ORF123 in vB_KpnM-VAC66), that are able to methylate the phage DNA to evade attacks of the bacterial restriction enzymes, involved in the restriction/methylation defense system [29]. These proteins play a role in the phage defense against their hosts, together with other mechanisms [30].

### 3.3. Phylogenetic Relationships

Twenty complete genomes of phages belonging to the *Tevenvirinae* subfamily were chosen to study their phylogenetic relationships; within these, genomes of phages infecting *Klebsiella, Cronobacter, Escherichia, Shigella, Erwinia, Enterobacter*, and *Vibrio* genera were chosen, with genomes ranging from 98,975 to 199,912 bp, in which 18 genomes were larger than 158 kbps. A multiple alignment for these nucleotide sequences was previously performed with the MAFFT software, then a phylogenetic tree on MEGA X software was generated using the maximum likelihood method (100 bootstrap replicates). The original tree is reported in Figure 4, where the sequences corresponding to vB_KpnM-VAC13 (in green) and vB_KpnM-VAC66 (dark red) are closely related, with a confidence level of 95%, and distant from other *Klebsiella* phages, such as MN013084, MN101229, MT701588, MN101225, or MW239157.

### 3.4. Different Features in vB_KpnM-VAC13 and vB_KpnM-VAC66 Genomes

#### 3.4.1. L-Shaped Tail Fiber

Interestingly, all the genetic sequences encoding morphogenesis-related proteins shared a nucleotidic identity of ≥95% (Figure 3b), except the L-shaped tail fiber proteins of both phages, which exhibited a low query cover (25%) and revealed an 83% nucleotidic identity. We also compared the amino-acid sequences and found a 77% query cover and a 50.97% identity. Therefore, we predicted the three-dimensional structure of each L-shaped tail fiber protein using the Phyre server (http://www.sbg.bio.ic.ac.uk/phyre2/html/page.cgi?id=index, accessed on 10 September 2021) and obtained very different model predictions, with confidences of 59.8% and 99.4% for vB_KpnM-VAC13 (Figure 5a) and vB_KpnM-VAC66 (Figure 5b), respectively. Furthermore, we aligned the amino-acid sequences of both L-shaped tail fiber proteins using the MUSCLE alignment tool (https://www.ebi.ac.uk/Tools/msa/muscle/, accessed on 9 September 2021) and realized that only the first 453 amino acid shared a 71% amino acid identity, whereas from this position onwards the sequences were extremely different, with 31% identity, 16% gaps, and a query cover of 66% (Figure 5c).

Using the conserved domains search tool of the NCBI (https://www.ncbi.nlm.nih.gov/Structure/cdd/wrpsb.cgi, accessed on 10 September 2021), we determined that the L-shaped tail fiber from vB_KpnM-VAC13 only carried one identified domain on the 868–1163 interval, which was the autotransporter adhesin Ag43 (E-value 8.66 × 10^−4^). By comparison, the L-shaped tail fiber from vB_KpnM-VAC66 showed a long-tail fiber domain (cI33689) on the 544–716 interval (E-value 3 × 10^−4^); a pyocin_knob domain (cd19958) on the 809-896 interval (E-value 1.12 × 10^−7^); and, finally, it also showed a conserved peptidase_S74 domain (pfam 13884) on the 1177–1229 interval (E-value 1.97 × 10^−14^), with a function of an intramolecular chaperone of endosialidase, with autoproteolytic activity [31,32]. These carboxy-terminal residues are thought to be auto-proteolytically removed after correct trimerization and folding [33].

#### 3.4.2. Homing Endonucleases

Interestingly, the comparison between both genomes revealed that vB_KpnM-VAC66 harbored in its genome nine more homing endonucleases than vB_KpnM-VAC13 (Figure 2 and Figure 3a and Table 2). However, the ORF268 of vB_KpnM-VAC66 may encode another homing endonuclease present in the genome of vB_KpnM-VAC66 and absent in vB_KpnM-VAC13, as it consists of a GIY-YIG domain-containing protein, and this domain is associated with homing endonucleases, which are present in a superfamily of enzymes that cleaves DNA [34,35].

Homing endonucleases are site-specific endonucleases that initiate mobility by introducing double-strand breaks (DSBs) at defined positions on DNA, triggering repair and recombination pathways that mobilize the endonuclease encoding region [35]. Mobility depends on host (RecA) or phage (UvsX) recombinase functions; in the case of these two lytic phages, they both encode a UvsX recombinase protein (ORFs 152 and 130 in vB_KpnM-VAC13 and VAC66, respectively), probably associated with other recombination processes independently of homing endonucleases in vB_KpnM-VAC13. Genomic data has revealed that homing endonucleases are extremely widespread in T-even-like phages [35].

An important characteristic of homing endonucleases in vB_KpnM-VAC66 is that most of their encoding genes were inserted near functionally critical bacteriophage genes, such as methyltransferases, DNA helicases, RNA ligases, or structural proteins. This is consistent with previous findings that report the presence of homing endonucleases within or near important genes, such as DNA polymerases [35]. Curiously, the homing endonuclease SegD, belonging to the GIY-YIG family, was found to be exactly in the same genetic environment in vB_KpnM-VAC66 as in T4 bacteriophage: between the major capsid protein (ORF29) and the capsid vertex protein (ORF31).

## 4. Discussion

In this work, we phenotypically characterized the lytic bacteriophage vB_KpnM-VAC66 and annotated its genome, compared it with the genome of another similar lytic bacteriophage, vB_KpnM-VAC13, and closely examined their differences.

Regarding the infection capacity of vB_KpnM-VAC66, we found that it efficiently infected its native host, K3320, in broth medium at MOIs of 0.1 and 1, of which the latter was the most efficient in decreasing the OD_600nm_ of the culture (Figure 1b); however, it also showed a good infection capacity in solid medium, with positive EOP values in nearly 21% of clinical isolates (Table 1). Indeed, differences in the host range of each bacteriophage were obtained when using different experimental approaches, because the spot test tends to overestimate the number of clinical strains of *K. pneumoniae* that were infected compared with the EOP, as previously claimed [36]. Furthermore, vB_KpnM-VAC66 showed a very short latent time with a medium burst size, two characteristics that are suitable for phage therapy [37,38]. However, the vB_KpnM-VAC13 showed a shorter time of adsorption (4 min), with a longer latent time (15 min) and a smaller burst size (4.83 PFU/mL) [12].

In addition to their differences in their host range or infectivity, both phages undoubtedly shared a large number of features: they belong to the *Myoviridae* family and *Tevenvirinae* subfamily, and are evolutionary correlated, as seen in the phylogenetic study, whereas other *K. pneumoniae*-infecting phages appear quite distant from them (Figure 4).

Considering the genomic aspect, both phages share the same genetic background (Figure 2 and Figure 3a). The most interesting findings were the divergence in the L-shaped tail fiber and the high predominance of homing endonucleases in the genome of vB_KpnM-VAC66. Indeed, vB_KpnM-VAC66 encodes ten of these enzymes that were absent in vB_KpnM-VAC13, and that are probably involved in horizontal gene transfer (HGT) and genetic mobility. This is similar to the findings described for the T4 phage, which encodes 15 homing endonucleases [35]. Endonucleases initiate nonreciprocal transfer of DNA segments containing their own gene, together with the flanking sequences, by cleaving the recipient DNA [34].

We also found a high diversity between both L-shaped tail fiber proteins (Figure 5). These proteins harbor a different RBD that may explain the fact that genomically similar phages can infect different clinical strains; furthermore, it has been claimed that bioengineering of this receptor modifies the host range of the bacteriophage [39], and that the enzymatic domain of the L-shaped tail fiber is subject to an intense HGT that allows the bacteriophage to rapidly shift its host spectrum without affecting the architecture of the receptor [40].

Among the main proteins that we annotated, we found many involved in morphogenesis and structure, ADP-ribosyltransferases, methyltransferases, lysis-related proteins and receptors, among others. We found several lysis-related proteins, such as two putative endolysins in each genome, which may constitute a potential strategy against multidrug resistant bacteria [24]. Endolysins have been recently bioengineered in combination with RBDs to increase their binding capacity and broaden their antimicrobial spectrum. In contrast to the lytic proteins, another interesting protein from the therapeutic perspective is the group of phage-borne depolymerases, which have been suggested as potential antivirulence agents, antibiotics adjuvants, and anti-biofilm molecules [41,42]. Within the L-shaped tail fiber protein of vB_KpnM-VAC66, we found an endosialidase domain, the S74_peptidase domain (pfam 13884), which is a type of depolymerase enzyme that this bacteriophage may use to hydrolyze the polysialic acid capsule of *K. pneumoniae.* This endosialidase is synthesized as a precursor of the C-terminal propeptide, which acts as a chaperone and is removed by a self-cleaving mechanism [31,32]. Consistent with the endosialidase domain found in the ORF79 tail fiber of vB_KpnM-VAC66, these depolymerases are predominantly located in RBPs structured as tail fibers [43]. Other works have focused on describing the enzymes that deploy degrading activity against specific *K. pneumoniae* capsular types [44].

Large phages may have arisen from short phages that acquired novel genetic functions, which probably conferred them some evolutionary advantages. It has also been described that large phages appeared to be much more efficient in the long-term maintenance in the bacterial population than short phages [45]. Even if their use in phage therapy needs to be further studied and validated, vB_KpnM-VAC13 has already shown good antibacterial activity both in vitro and in an in vivo model combined with imipenem and the repurposed drug mitomycin C [12]. Fascinatingly, many of their genetic functions are unidentified, unexploited, and unknown, which underpins the need to deeply study phages to enlarge the genetic diversity of these viruses, the hallmark of the phage universe.

## 5. Conclusions

In this study, we compared the genomes of two lytic phages that infect clinical isolates of *K. pneumoniae*. The *Klebsiella* phage vB_KpnM-VAC66 was phenotypically characterized in this work. We found that vB_KpnM-VAC66 exhibits a short latency time and a medium burst size. Regarding its genome, we observed a high predominance of homing endonucleases compared to vB_KpnM-VAC13, as it was observed that the former encoded ten different endonucleases. Furthermore, we realized that their L-shaped tail fibers were also highly dissimilar, which may explain their different host range; interestingly, the C-terminal region of the L-shaped tail fiber of vB_KpnM-VAC66 encoded an endosialidase domain, acting as a depolymerase domain that hydrolyzes the capsule of some *K. pneumoniae* clinical isolates. Therefore, our results confirm the important role of the L-shaped tail fiber proteins and homing endonucleases in the lytic activity of phages, leading to interesting biotechnological applications.

## Figures and Tables

**Figure 1 viruses-14-00006-f001:**
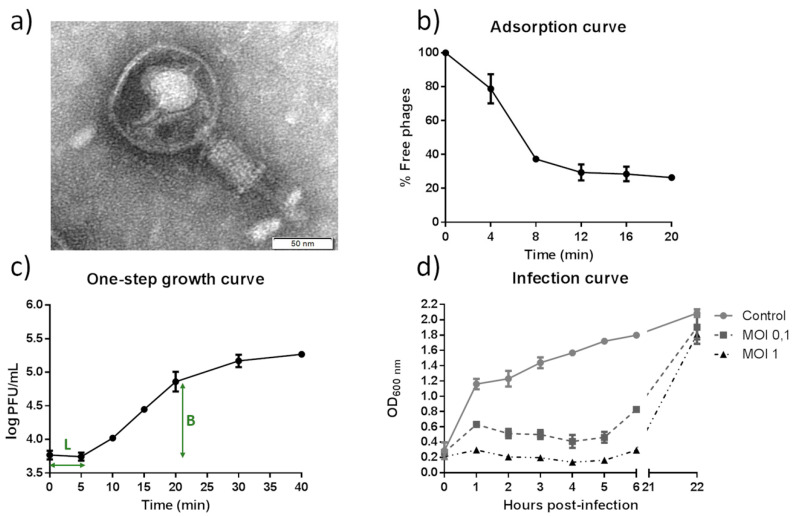
Phenotypic characterization of vB_KpnM-VAC66. (**a**) TEM photography. (**b**) Adsorption curve using the clinical strain K3320. (**c**) One-step growth curve using the clinical strain K3320; L stands for latency period and B for burst size. (**d**) Infection curve of K3320 at a MOIs of 0.1 (dark grey squares) and 1 (black triangles); light grey circles represent the growth of K3320 without infection (control).

**Figure 2 viruses-14-00006-f002:**
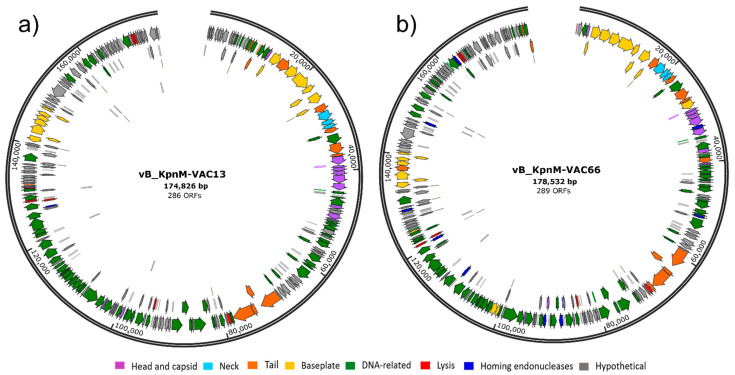
Circular representations of vB_KpnM-VAC13 (**a**) and vB_KpnM-VAC66 (**b**) genomes, created with SnapGene^®^ 5.3.2.

**Figure 3 viruses-14-00006-f003:**
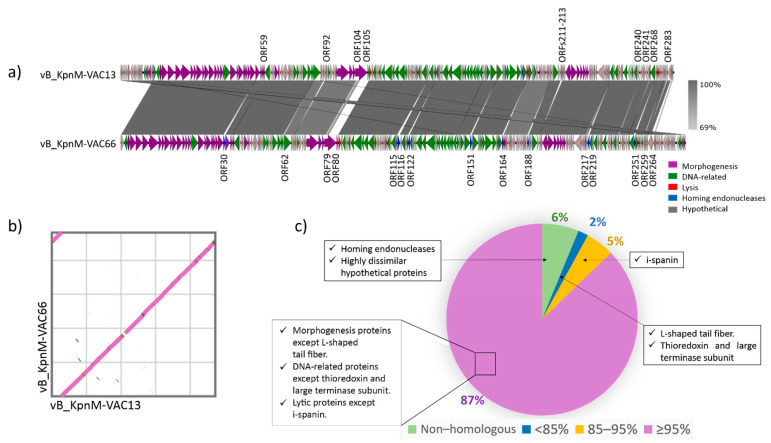
(**a**) Comparison between genomes of phages vB_KpnM-VAC13 and vB_KpnM-VAC66, plotted with EasyFig_2.2.5. (**b**) Dot plot comparison between both genomes, vB_KpnM-VAC13 (X axis) and vB_KpnM-VAC66 (Y axis) using the ViPTree online tool. (**c**) Percentage identity between all the nucleotidic sequences in both genomes, assessed using the blastn tool (https://blast.ncbi.nlm.nih.gov/Blast.cgi?PAGE_TYPE=BlastSearch, accessed on 18 September 2021).

**Figure 4 viruses-14-00006-f004:**
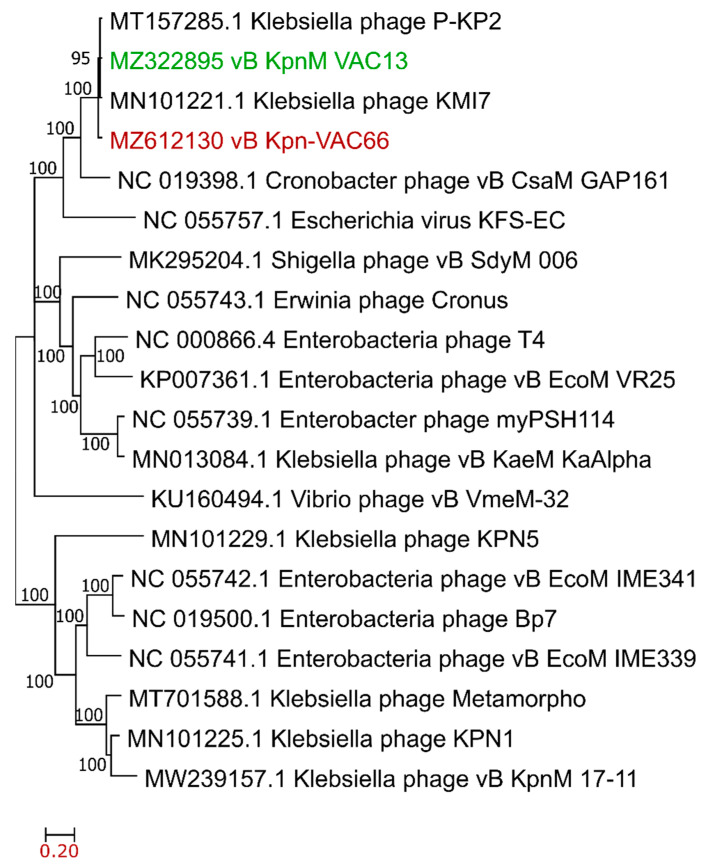
Phylogenetic tree built using the maximum likelihood method (MEGA X) with a bootstrap of 100. The numbers provide confidence levels for each clade.

**Figure 5 viruses-14-00006-f005:**
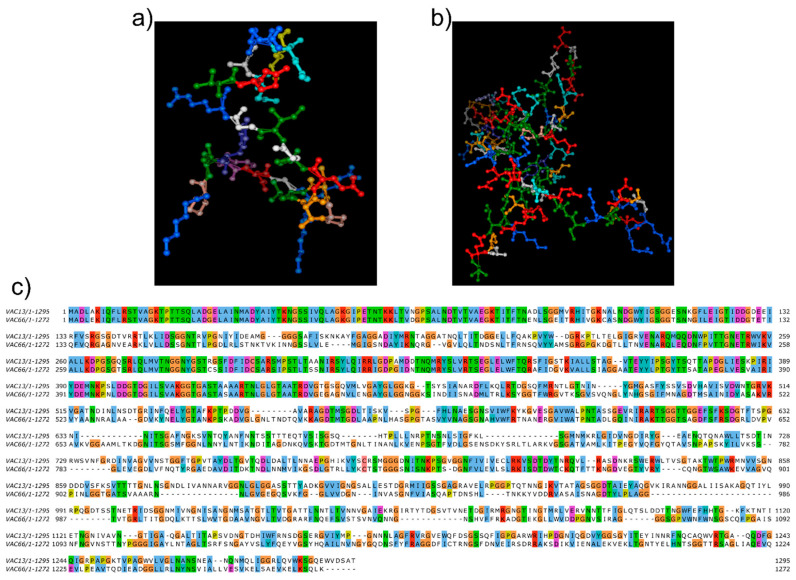
In silico predictions of the three-dimensional models of L-shaped tail fibers for vB_KpnM-VAC13 (**a**) and vB_KpnM-VAC66 (**b**), using Phyre. (**c**) Alignment between the amino acid sequences of L-shaped tail fiber protein performed with MUSCLE and colored by group of amino acids.

**Table 1 viruses-14-00006-t001:** Host range of vB_KpnM-VAC13 and vB_KpnM-VAC66 using 48 clinical isolates of *K. pneumoniae*. ++: clear lysis plaque; +: semi-clear lysis plaque with some resistance; + −: semi-turbid lysis plaque with more resistance; −: no lysis plaque. The values of EOP are reported in brackets.

Strain	Spot Test and EOP vB_KpnM-VAC13	Spot Test and EOP vB_KpnM-VAC66	Strain	Spot Test and EOP ^1^ vB_KpnM-VAC13	Spot Test and EOP vB_KpnM-VAC66
SCISP4C	−	−	K2783	+ −	−
SCISP2C	−	+ − (0.032)	K2715	−	+ − (0.023)
K3667	−	−	K2707	−	−
K3579	+ −	−	K2691	−	−
K3575	−	+ −	K2597	−	+ −
K3574	+ − (0.328)	+ − (0.0031)	K2551	+ (0.012)	+ −
K3573	+ − (0.17)	+ − (0.0026)	K2535	−	+ −
K3571	+ −	−	ST974-OXA48	+ −	+ (0.044)
K3509	+ − (0.03)	−	ST899-OXA48	+ −	−
K3416	+ −	+ −	ST258-KPC3	−	−
K3325	+ −	−	ST11-OXA48	−	−
K3324	+ − (0.77)	+ (0.266)	ST15-OXA48	+ −	+ −
K3323	+ −	−	ST512-KPC3	−	−
K3322	−	−	ST13-OXA48	+ (0.094)	−
K3321	+ − (0.328)	+ − (0.468)	ST340-VIM1	−	−
K3320	+ − (0.458)	+ (1)	ST846-OXA48	+ −	−
K3318	−	+ −	ST11-VIM1	−	+ − (0.037)
K2990	+ −	−	ST147-VIM1	+ −	+ −
K2989	−	+ −	ST101-KPC2	−	−
K2986	+ −	−	ST16-OXA48	+	−
K2984	−	+ − (0.046)	ST437-OXA245	++ (0.48)	−
K2983	+ − (0.8)	−	ST11-OXA245	+ −	− (0.0067)
K2982	−	−	ST15-VIM1	+ −	+ −
K2791	−	−	ST405-OXA48	−	+ − (0.038)

^1^ The natural host for vB_KpnM-VAC13 was the reference strain *K. pneumoniae* ATCC^®^10031^TM^ (EOP value = 1).

**Table 2 viruses-14-00006-t002:** Genomic differences between both phages, vB_KpnM-VAC13 and vB_KpnM-VAC66. On top, the ORFs present in vB_KpnM-VAC166 and absent in vB_KpnM-VAC13; on the right, the ORFs present in vB_KpnM-VAC13 and absent in vB_KpnM-VAC66.

ORF	Present in vB_KpnM-VAC66 and Absent in vB_KpnM-VAC13
30	SegD homing endonuclease
62	Hypothetical protein
80	Distal long tail fiber assembly catalyst
115	Homing Endonuclease
116	Homing Endonuclease
122	Homing Endonuclease
151	Homing Endonuclease
164	Homing Endonuclease
188	Homing Endonuclease
207	Hypothetical protein
219	Homing Endonuclease
259	Homing Endonuclease
264	Hypothetical protein
268	GIY-YIG domain containing Protein
283	Hypothetical protein
ORF	Present in vB_KpnM-VAC13 and absent in vB_KpnM-VAC66
59	Hypothetical protein
92	Hypothetical protein
105	Single-stranded DNA-binding protein
211–213	Three hypothetical proteins
227–228	Two hypothetical proteins

## Data Availability

The datasets of the current study are the two genomes whose accession numbers are MZ322895 (vB_KpnM-VAC13) and MZ612130 (vB_KpnM-VAC66), included in the BioProject PRJNA739095 (https://www.ncbi.nlm.nih.gov/bioproject/?term=PRJNA739095 accessed on 1 February 2021).

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
