# Peer review of "Phenotypic and Genomic Comparison of Klebsiella pneumoniae Lytic Phages: vB_KpnM-VAC66 and vB_KpnM-VAC13"

_viruses, 2021, doi:10.3390/v14010006_

Round 1
Reviewer 1 Report
The manuscript “Phenotypic and genomic comparison…” by Pacios et al deals with the DNA comparison of two lytic phages that infect Klebsiella pneumoniae belonging to Myoviridae family, Tevenvirinae subfamily. In fact, one of them (vB_KpnM-VAC13) was already phenotypically characterized and published by the same coauthors in this year 2021. Here, the main phenotypic characteristics of the second phage (vB_KpnM-VAC66) have been studied and compared between them.
In principle, two almost identical phages that share many phenotypic properties and reaching a 97,56% nucleotide similarity, should not arouse special scientific interest. Nevertheless, some relevant differences between such phages merit this particular study, although it is also clear that most particular aspects of vB_KpnM-VAC66 phage remain unanswered. In summary, the data provided in this work are sufficiently interesting but need to be revised on some areas.
Major points
- The first comment is about the purification of phages. According to the explanations in Materials and Methods (section 2.2), “The plaque-purified phages were amplified and propagated in liquid media, infecting their natural hosts when they reached an OD600 nm=0.4. Lysed bacteria were removed by centrifugation and supernatants were stored at 4°C”. Does it mean that phages are not further purified, normally through a CsCl gradient step? This protocol, or something similar, is practically mandatory to obtain highly purified phages with the highest possible titer. In addition, how long are the phage lysates stored at 4ºC? What is the “liquid media” of this storage? Is there any preservative compound? Have the authors checked the evolution of phage titer over time at 4ºC, -20ºC, and -80ºC with glycerol? Does this phage retain its titer with lyophilization and further resuspension? These details are important for the study of any new phage and are lacking in the manuscript.
- Several comments about the lysis-related proteins:
- The statement that holins form “large holes” in the cytoplasmic membrane is rather ambiguous, since there is no correlation or comparison with an estimation of their size, a subject extensively studied by several articles of Ry Young’s group. Here, it would be enough the comment that “holins presumably oligomerize to form holes in the cytoplasmic membrane to allow the passage of endolysins”.
- The sentence “one endolysin predicted encoded the domain PET-M15-4, also described as D-alanyl-D-alanine carboxypeptidase, that displays L-Ala-D-Glu activity…” is confusing. A “carboxypeptidase” is one catalytic activity and “L-D-endopeptidase” is another. Please, clarify which activity the predicted domain refers to.
- Figure 1d depicts the infection curve of K3320 with phage vB_KpnM-VAC66 at two different MOIs, but prolonging the incubation time beyond the standard lysis, which is not usually shown. This allows the resuming of growth due to resistants. Have the authors quantified this mutation rate and compared with those described to other similar phages? These data should be of interest in case of the potential therapeutic use of these phages, as discussed in the manuscript.
- In the Abstract section, line 33, a necessary clarification of the definition of “homology”: this concept is not associated with a numerical percentage, which is only linked to “similarity” (nucleotide or amino acid). Thus, the phages described here are homologous, with a 97,56% nucleotide similarity between them. Please, correct the sentence.
- The exact size of the genomes of both phages studied in this work should be also included in the Introduction section, and not only in the Results (section 3.2, line 224).
- There is discordance in the qualification of “burst size” of the phage vB_KpnM-VAC66: in line 395 it is stated as “medium” and in line 450 as “high”. Please, unify the criteria.
- It is rather curious that authors comment as “astonishing” the location of homing endonucleases in vB_KpnM-VAC66, since “most of their encoding genes were inserted near functionally critical bacteriophage genes” (line 375). But just in the line below, it is stated “This is consistent with previous findings that report the presence of homing endonucleases within or near important genes”.
Minor points
- In the title, Klebsiella pneumoniae must be in italics.
- In the #2 affiliation of coauthors, the correct sentence is “on behalf of…”.
- In the abstract, TEM must be defined.
- Line 80: Correct to “the phenotypic and genomic analyses…” or “the phenotypic and genomic analysis that we report here constitutes…”.
- Line 92: This time, Klebsiella pneumoniae should not be written in italics.
- Line 122: Correct “phages” to “phage”.
- Line 131: Correct “this bacteriophage, that is…” to “this bacteriophage, which is…”.
- Line 188: Correct “CaCl2 1 mM” to “1 mM CaCl2”.
- Line 359; Correct “nine more homing endonucleases that…” to “nine more homing endonucleases than…”.
- Line 457: Correct “rle” to “role”.
- Line 197: Delete “efficiency of plating” since it has previously defined in line 111.
- Line 225: The acronym CDS (supposedly “coding sequences”) is not defined and in this line is the first time used.
Author Response
The manuscript “Phenotypic and genomic comparison…” by Pacios et al deals with the DNA comparison of two lytic phages that infect Klebsiella pneumoniae belonging to Myoviridae family, Tevenvirinae subfamily. In fact, one of them (vB_KpnM-VAC13) was already phenotypically characterized and published by the same coauthors in this year 2021. Here, the main phenotypic characteristics of the second phage (vB_KpnM-VAC66) have been studied and compared between them.
In principle, two almost identical phages that share many phenotypic properties and reaching a 97,56% nucleotide similarity, should not arouse special scientific interest. Nevertheless, some relevant differences between such phages merit this particular study, although it is also clear that most particular aspects of vB_KpnM-VAC66 phage remain unanswered. In summary, the data provided in this work are sufficiently interesting but need to be revised on some areas.
Major points
- The first comment is about the purification of phages. According to the explanations in Materials and Methods (section 2.2), “The plaque-purified phages were amplified and propagated in liquid media, infecting their natural hosts when they reached an OD600 nm=0.4. Lysed bacteria were removed by centrifugation and supernatants were stored at 4°C”. Does it mean that phages are not further purified, normally through a CsCl gradient step? This protocol, or something similar, is practically mandatory to obtain highly purified phages with the highest possible titer. Unfortunately, at our laboratory we do not have the necessary equipment to perform CsCl purification.
In addition, how long are the phage lysates stored at 4ºC?
What is the “liquid media” of this storage? Is there any preservative compound?
It is SM Buffer (100 mM NaCl, 10 mM MgSO4, 20 mM Tris-HCl pH 7.5) without any preservative compound.
Have the authors checked the evolution of phage titer over time at 4ºC, -20ºC, and -80ºC with glycerol?
We have checked the evolution of phage titer at 4ºC over a year for both vB_KpnM-VAC13 and vB_KpnM-VAC66 and we observed a reduction of 47% for vB_KpnM-VAC13 and no reduction at all for vB_KpnM-VAC66. We have included this in lines 95 and 105-108.
Does this phage retain its titer with lyophilization and further resuspension? These details are important for the study of any new phage and are lacking in the manuscript.
Again, it is not possible to do the lyophilization step at our laboratory because we do not have the equipment.
- Several comments about the lysis-related proteins:
- The statement that holins form “large holes” in the cytoplasmic membrane is rather ambiguous, since there is no correlation or comparison with an estimation of their size, a subject extensively studied by several articles of Ry Young’s group. Here, it would be enough the comment that “holins presumably oligomerize to form holes in the cytoplasmic membrane to allow the passage of endolysins”. It has been rephrased (lines 316-318)
- The sentence “one endolysin predicted encoded the domain PET-M15-4, also described as D-alanyl-D-alanine carboxypeptidase, that displays L-Ala-D-Glu activity…” is confusing. A “carboxypeptidase” is one catalytic activity and “L-D-endopeptidase” is another. Please, clarify which activity the predicted domain refers to.
We have now stated that PET-M15-4 is a domain with endopeptidase activity (line 322). Based on what has been already claimed by Oliveira H et al.: “PET-M15-4 is a domain also described as d-alanyl-d-alanine carboxypeptidase; however, it has been predicted to display l-Ala-d-Glu activity”. Oliveira, H., Melo, L. D., Santos, S. B., Nóbrega, F. L., Ferreira, E. C., Cerca, N., Azeredo, J., & Kluskens, L. D. (2013). Molecular aspects and comparative genomics of bacteriophage endolysins. Journal of virology, 87(8), 4558–4570. https://doi.org/10.1128/JVI.03277-12.
- Figure 1d depicts the infection curve of K3320 with phage vB_KpnM-VAC66 at two different MOIs, but prolonging the incubation time beyond the standard lysis, which is not usually shown. This allows the resuming of growth due to resistants. Have the authors quantified this mutation rate and compared with those described to other similar phages? These data should be of interest in case of the potential therapeutic use of these phages, as discussed in the manuscript.
The mutation rate of vB_KpnM-VAC13 was already calculated in the following article: Enhanced antibacterial activity of repurposed mitomycin C and imipenem in combination with the lytic phage vB_KpnM-VAC13 against clinical isolates of Klebsiella pneumoniae. Pacios O, Fernández-García L, Bleriot I, Blasco L, González-Bardanca M, López M, Fernández-Cuenca F, Oteo J, Pascual Á,Martínez-Martínez L, Domingo-Calap P, Bou G, Tomás M. Antimicrob Agents Chemothery (2021). 65:e00900-21. https://doi.org/10.1128/AAC.00900-21.
The mutation rate of vB_KpnM-VAC66 has been assessed now and found to be 5·10-2 ± 0.03. Both mutation rates are now included in the manuscript (lines 158-166 for material and methods, and 239-242 for results).
- In the Abstract section, line 33, a necessary clarification of the definition of “homology”: this concept is not associated with a numerical percentage, which is only linked to “similarity” (nucleotide or amino acid). Thus, the phages described here are homologous, with a 97,56% nucleotide similarity between them. Please, correct the sentence.
It has been corrected as suggested (line 33). Moreover, the numerical percentages that were wrongly associated with the homology term have now been associated with the identity percentage that the blast tool indicates (lines 264, 268, 269, 367 and 368), following the indications of this article: Homology, similarity, and identity in peptide epitope immunodefinition. Kanduc, D. (2012), J. Pept. Sci., 18: 487-494. https://doi.org/10.1002/psc.2419
- The exact size of the genomes of both phages studied in this work should be also included in the Introduction section, and not only in the Results (section 3.2, line 224).
The size of the genomes has been included in the Introduction (lines 74-75)
- There is discordance in the qualification of “burst size” of the phage vB_KpnM-VAC66: in line 395 it is stated as “medium” and in line 450 as “high”. Please, unify the criteria.
It has been unified as “medium” (line 478)
- It is rather curious that authors comment as “astonishing” the location of homing endonucleases in vB_KpnM-VAC66, since “most of their encoding genes were inserted near functionally critical bacteriophage genes” (line 375). But just in the line below, it is stated “This is consistent with previous findings that report the presence of homing endonucleases within or near important genes”.
True, the word “astonishing” has been removed (line 402)
Minor points
- In the title, Klebsiella pneumoniae must be in italics. Done (line 2)
- In the #2 affiliation of coauthors, the correct sentence is “on behalf of…”. Done (line 11)
- In the abstract, TEM must be defined. Done (lines 30-31)
- Line 80: Correct to “the phenotypic and genomic analyses…” or “the phenotypic and genomic analysis that we report here constitutes…”. Done (line 82)
- Line 92: This time, Klebsiella pneumoniae should not be written in italics. Done (line 94)
- Line 122: Correct “phages” to “phage”. Done (line 128)
- Line 131: Correct “this bacteriophage, that is…” to “this bacteriophage, which is…”. Done (line 137)
- Line 188: Correct “CaCl2 1 mM” to “1 mM CaCl2”. Done (line 204)
- Line 359; Correct “nine more homing endonucleases that…” to “nine more homing endonucleases than…”. Done (line 387)
- Line 457: Correct “rle” to “role”. Done (line 485)
- Line 197: Delete “efficiency of plating” since it has previously defined in line 111. Done (line 214)
- Line 225: The acronym CDS (supposedly “coding sequences”) is not defined and in this line is the first time used. Done (line 247)

Reviewer 2 Report
The manuscript gives the sequence of a couple more T4-like phages, presenting them as potential for use in phage therapy of K. pneumoniae. They give some comparative sequence data. They give the growth properties of one of them. It's done well enough to give a general idea of what the phages are. The findings that there are differences in homing nucleases and in the tail fiber between the two is typical of phages this closely related.
In the phenotypic characterization absorption is better measured by a rate constant than a time, since the time is dependent on the bacterial cell concentration. Lag time and burst size are better measured by severe dilution to stop absorption rather than all this washing, because it's hard to know how the time of washing plays into the lag time. Similarly without dilution to prevent reinfection, you end up picking some arbitrary time to measure the burst size before all the cells burst, because the curve never really plateaus.
I think the axis on fig. 1c is supposed to be log pfu/ml.
The discussion about 50% hypothetical proteins is nonsense. The only difference between most of the "hypothetical proteins" and the other proteins is that the other proteins matched something in T4, where they bothered to give it a name. The ones labeled "hypothetical proteins" match protein families in bunches of other T4-like phages that no one bothered to name or characterize.
The discussion about the number of proteins in different divergence classes is more useful. I suspect that the class called "highly diverged" is really "proteins with no similarity in the other phage". "Highly diverged" would mean that there is similarity, but it's low.
Fig. 2. These genomes are circularly permuted, so the ends chosen for presentation are arbitrary. The crossing pattern in Fig2a is because different arbitrary ends were chosen in the two genomes before making this figure.
Table 2 is unreadable: For example, ORF30 is said to be ND (not determined?) in VAC13, and a homing nuclease in VAC66. What is actually true is that ORF30 in VAC66 is a homing nuclease, and the syntenous position in VAC13 is between ORF54 and ORF55 and is not occupied by a reading frame. This table needs to be explained better. Also, since it's to be read in three columns, drop vertical lines or do something to demark the folded columns.
Author Response
The manuscript gives the sequence of a couple more T4-like phages, presenting them as potential for use in phage therapy of K. pneumoniae. They give some comparative sequence data. They give the growth properties of one of them. It's done well enough to give a general idea of what the phages are. The findings that there are differences in homing nucleases and in the tail fiber between the two is typical of phages this closely related.
In the phenotypic characterization absorption is better measured by a rate constant than a time, since the time is dependent on the bacterial cell concentration.
Lag time and burst size are better measured by severe dilution to stop absorption rather than all this washing, because it's hard to know how the time of washing plays into the lag time. Similarly without dilution to prevent reinfection, you end up picking some arbitrary time to measure the burst size before all the cells burst, because the curve never really plateaus. We have based our protocols of the adsorption and one-step growth curves on published articles where new bacteriophages are characterized, such as: “Characterization of newly isolated lytic bacteriophages active against Acinetobacter baumannii”. Merabishvili, M., Vandenheuvel, D., Kropinski, A. M., Mast, J., De Vos, D., Verbeken, G., Noben, J. P., Lavigne, R., Vaneechoutte, M., & Pirnay, J. P. (2014). PloS one, 9(8), e104853. https://doi.org/10.1371/journal.pone.0104853; and “Combined Use of the Ab105-2φΔCI Lytic Mutant Phage and Different Antibiotics in Clinical Isolates of Multi-Resistant Acinetobacter baumannii”. Blasco, L., Ambroa, A., Lopez, M., Fernandez-Garcia, L., Bleriot, I., Trastoy, R., Ramos-Vivas, J., Coenye, T., Fernandez-Cuenca, F., Vila, J., Martinez-Martinez, L., Rodriguez-Baño, J., Pascual, A., Cisneros, J. M., Pachon, J., Bou, G., & Tomas, M. (2019). Microorganisms, 7(11), 556. https://doi.org/10.3390/microorganisms7110556
I think the axis on fig. 1c is supposed to be log pfu/ml. The title of the Y axis on Fig.1C has been changed by log PFU/mL.
The discussion about 50% hypothetical proteins is nonsense. The only difference between most of the "hypothetical proteins" and the other proteins is that the other proteins matched something in T4, where they bothered to give it a name. The ones labeled "hypothetical proteins" match protein families in bunches of other T4-like phages that no one bothered to name or characterize. The discussion about the hypothetical proteins has been removed (lines 247-250)
The discussion about the number of proteins in different divergence classes is more useful. I suspect that the class called "highly diverged" is really "proteins with no similarity in the other phage". "Highly diverged" would mean that there is similarity, but it's low.
“Highly diverged” has been replaced by “highly dissimilar”, as suggested (lines 38, 481 and Fig 3.c). Similarly, in line 271 the term “diverged” has been replaced by “differed”, to avoid any ambiguous interpretation.
Fig. 2. These genomes are circularly permuted, so the ends chosen for presentation are arbitrary. The crossing pattern in Fig2a is because different arbitrary ends were chosen in the two genomes before making this figure.
Yes, that is true. Indeed, we decided to keep the original FASTA sequences that we obtained after sequencing to annotate both genomes and represent them in Fig 2 and Fig 3a.
Table 2 is unreadable: For example, ORF30 is said to be ND (not determined?) in VAC13, and a homing nuclease in VAC66. What is actually true is that ORF30 in VAC66 is a homing nuclease, and the syntenous position in VAC13 is between ORF54 and ORF55 and is not occupied by a reading frame. This table needs to be explained better. Also, since it's to be read in three columns, drop vertical lines or do something to demark the folded columns.
Table 2 has been split into two different tables where the genomic differences between both bacteriophages are indicated (lines 257-259); on the left, the ORFs present in vB_KpnM-VAC13 but absent in vB_KpnM-VAC66 are reported, whereas on the right we have now collected the ORFs present in vB_KpnM-VAC66 but absent in the other bacteriophage. We hope this is clearer now.

Reviewer 3 Report
Attached

Author Response
The authors annotated and compared the genomes of two lytic phages, namely vB_KpnM-VAC13 and vB_KpnM-VAC66 that infect clinical strains of K. pneumoniae and phenotypically characterized vB_KpnM-VAC66. The findings in the manuscript demonstrate that these viruses could be useful in the development of alternative therapies against MDROs to tackle the problem of AMR. I hope I helped the authors improve the manuscript. I have a few minor comments:
Line 90 37ºC should be 37ᵒC. Done (line 91)
Line 212,214 Standardize hours to h. Done (lines 228 and 230)
Line 235 Rearrange Table 2 such that headings are in the same page Table 2 has been rearranged, split into two to clarify the information and in the same page as the headings (lines 257-259).
Line 450 Please rephrase the sentence "Regarding its genome, we observed a high predominance of homing endonucleases compared to vB_KpnM-VAC13 was observed the former encoding ten different endonucleases." It has been rephrased (lines 479-480)
Line 456 Please correct "the importante rle". Done (line 485)
